

# Above Cloud CCN Concentrations Help to Sustain Some Arctic Low-Level Clouds

Lucas J. Sterzinger[1,2,3] and Adele L. Igel[1]

[1]Department of Land, Air and Water Resources, University of California, Davis, Davis, California, USA
[2]NASA Goddard Space Flight Center, Bethesda, Maryland, USA
[3]Adnet Systems, Inc, Rockridge, Maryland, USA

**Correspondence:** Adele L. Igel (aigel@ucdavis.edu)

**Abstract.** Recent studies have reported observations of enhanced aerosol concentrations directly above the Arctic boundary layer, and it has been suggested that Arctic boundary layer clouds could entrain these aerosol and activate them. We use an idealized LES modeling framework where aerosol concentrations are kept low in the boundary layer, and increased up to 50x in the free troposphere. We find that the simulations with higher tropospheric aerosol concentrations persisted for longer and

had higher liquid water path. This is due to direct entrainment of the tropospheric aerosol into the cloud layer which results in a precipitation suppression from the increase in cloud droplet number and in stronger radiative cooling at cloud top due to the higher liquid water content at cloud top, which causes stronger circulations maintaining the cloud in the absence of surface forcing. Together, these two responses result in a more well-mixed boundary layer with a top that does not move rapidly in time such that it remains in contact with the tropospheric aerosol reservoir and can maintain entrainment of those aerosol

particles. The boundary layer aerosol and cloud droplet concentrations, however, remained low in all simulations. Surface based measurements in this case would not necessarily suggest the influence of tropospheric aerosol on the cloud, despite it being necessary for stable cloud persistence.

## 1   Introduction

The Arctic is now estimated to be warming at four times the global mean warming rate (Rantanen et al., 2022). Clouds play a

large role in this amplification, with the net cloud feedback in the Arctic estimated to be +0.58 K with a doubling of $CO_2$, which contributes to 15% of the warming in the Arctic in such a scenario (Taylor et al., 2013). Low level mixed-phase clouds are crucial regulators of Arctic climate (Intrieri et al., 2002; Shupe and Intrieri, 2004; Sedlar et al., 2011) and are ubiquitous (Shupe et al., 2006, 2011; Shupe, 2011). These clouds' precise radiative forcing at the surface is not well quantified; for a majority of the year they exert a warming effect on the surface due to the high albedo of an ice surface and limited solar radiation (Shupe

and Intrieri, 2004; Sedlar et al., 2011). During the late summer, however, the clouds can have a cooling effect as surface albedo decreases due to melting ice and solar insolation increases. Properly modeling these clouds is crucial to accurately projecting Arctic and global climate change, yet representation of Arctic low-level clouds in models has remained a challenge. (Klein et al., 2009; Morrison et al., 2009, 2011, 2012; Sotiropoulou et al., 2016).



Low-level Arctic clouds have been observed to exist for days at a time (Shupe, 2011; Shupe et al., 2011; Morrison et al.,
2012; Verlinde et al., 2007). This is especially curious given the low aerosol concentrations in the Arctic; boundary layer aerosol
concentrations are at a minimum in the summer (Mauritsen et al., 2011; Heintzenberg et al., 2015) with typical values less than
100 cm$^{-3}$ and sometimes less than 1 cm$^{-3}$. Such low concentrations may be insufficient to maintain clouds (Mauritsen et al.,
2011; Stevens et al., 2018; Sterzinger et al., 2022). However, measurements taken at the surface may not be representative of
the rest of the lower atmosphere. Aerosol concentrations have been observed to be higher in the free troposphere (FT) than
in the boundary layer (BL) (Lonardi et al., 2022; Creamean et al., 2021; Wylie and Hudson, 2002; Hegg et al., 1995; Igel
et al., 2017). More specifically, using tethered balloon data from Oliktok Pt, Alaska spanning late spring 2017 through early
fall 2018, Creamean et al. (2021) found that above cloud aerosol concentrations were higher than those below cloud in 38%
of profiles analyzed. Lonardi et al. (2022) and Igel et al. (2017), using summertime data from the high Arctic, similarly found
higher concentrations of tropospheric aerosol concentrations when compared to the surface, but these studies presented data
from a limited number of days.

Here we extend the analysis presented by Lonardi et al. (2022) to include all tethered balloon profiles from the high Arctic
collected during MOSAiC (Shupe et al., 2022) with a well-defined temperature inversion to mark the transition to the free
troposphere that is at least 100m below the profile top (Pilz et al., 2022). Figure 1 shows vertical profiles of potential tem-
perature and aerosol concentration for particle diameters >12 nm from these days. All aerosol profiles (Fig. 1b) have higher
concentrations above the inversion than at any level below the inversion with the exception of 24 July 2020. Some profiles
show free-tropospheric aerosol concentrations in the low 100s cm$^{-3}$, while others are seen to reach 1000 cm$^{-3}$ or more. In all
profiles, near-surface aerosol concentrations were quite low, most below 200 cm$^{-3}$ and some well below 100 cm$^{-3}$, despite
the higher concentrations in the free troposphere.

Igel et al. (2017) found that entrainment of such elevated concentrations of aerosol particles above the inversion can be an
important source of aerosol for the Arctic boundary layer. Shupe et al. (2013) found that the large aerosol particles needed to
form and sustain Arctic stratocumulus were predominantly advected from lower latitudes. It may be that with too few aerosol
in the boundary layer, entrainment of aerosol from the troposphere is necessary to sustain clouds for the duration observed in
studies such as Shupe et al. (2011) and Morrison et al. (2012). Without such a source of aerosol, clouds may exist in a tenuous
regime and further dissipate (Sterzinger et al., 2022; Mauritsen et al., 2011). At the other end of the cloud lifetime, (Silber
et al., 2020) found that the presence of aerosol may be important for the transition of thin nonturbulent clouds to thick, high
liquid water clouds.

Many modeling studies of Arctic cloud-aerosol processes (e.g. Sterzinger et al., 2022; Stevens et al., 2018) rely on near-
surface measurements of aerosol concentrations to initialize concentrations throughout the entire domain. Given 1) the decou-
pling from the surface so often seen in the Arctic boundary layer and 2) a more polluted troposphere being a potential source
of aerosol for Arctic boundary layer clouds, it's likely that these surface measurements are not always representative of the
aerosol concentrations influencing the cloud layer (Igel et al., 2017).

In this study, we use idealized modeling to investigate the sensitivity of Arctic mixed-phase boundary layer clouds to in-
creased concentrations in tropospheric aerosol - specifically aerosol that can act as cloud condensation nuclei. We present a





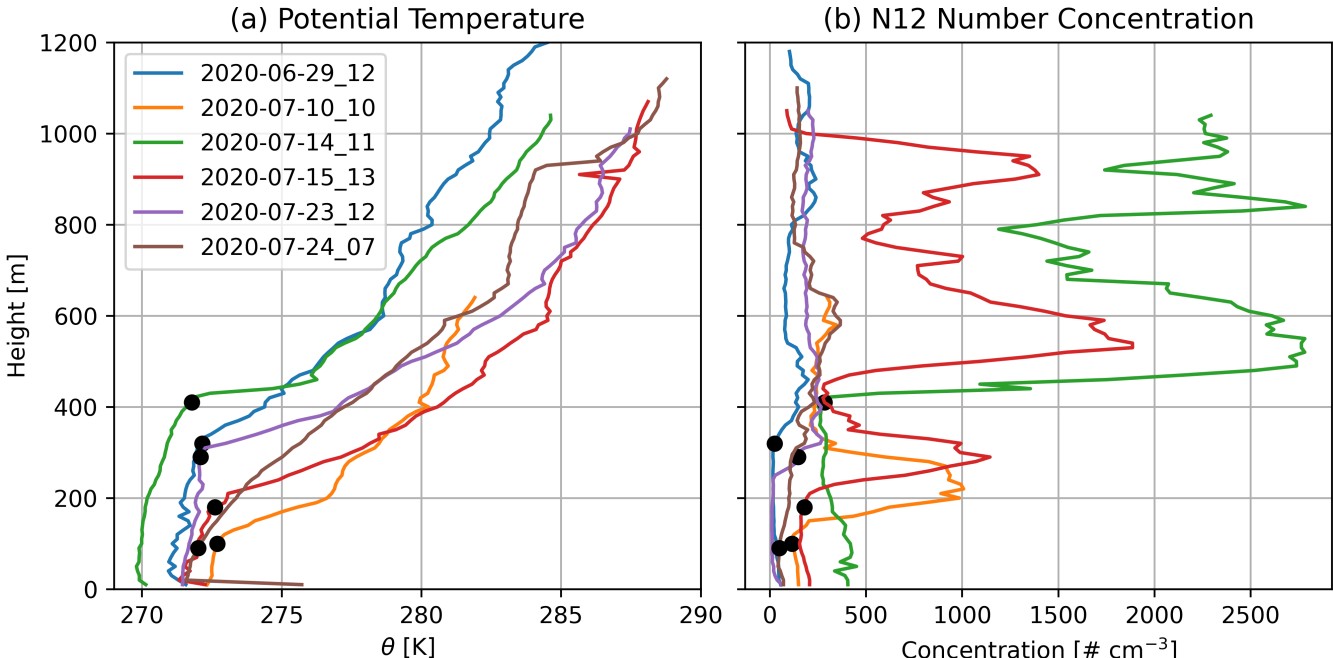

**Figure 1.** (a) Potential temperature and (b) aerosol concentration (>12 nm) for select profiles during the MOSAiC campaign. Black dots represent the top of the boundary layer for each profile.

suite of simulations, each with different tropospheric aerosol concentrations and examine the effect of these varied concentra-

tions on aerosol, cloud, and boundary layer properties.

## 2   Methodology

### 2.1   Model and Simulation Setup

We used the Colorado State University Regional Atmospheric Modeling System (RAMS; Cotton et al., 2003) to run large eddy simulations, a scale at which RAMS has been used successfully in prior studies (e.g. Cotton et al., 1992; Jiang et al., 2001;

Jiang and Feingold, 2006; Sokolowsky et al., 2022) and has proven to be insightful in studying aerosol-cloud interactions in Arctic clouds in similar LES setups (Bulatovic et al., 2021; Sterzinger et al., 2022).

RAMS uses a double-moment bulk microphysics scheme (Walko et al., 1995; Meyers et al., 1997; Saleeby and Cotton, 2004) predicting hydrometeor mass and number concentrations for cloud, rain, ice, snow, aggregates, graupel, and hail. The scheme includes a prognostic aerosol treatment (Saleeby and van den Heever, 2013) which tracks aerosol mass and number as

well as accounting for removal by hydrometeor formation and regeneration by hydrometeor evaporation. Cloud droplets are activated from aerosol particles using Köhler theory by referencing lookup tables (Saleeby and Cotton, 2004) and hydrometeor diffusional growth is explicitly dependent on supersaturation. Ice nucleation is parameterized following DeMott et al. (2010) as



described in Saleeby and van den Heever (2013). Both CCN and INP are returned to the atmosphere upon complete evaporation of liquid drops and complete sublimation of ice particles, respectively. Secondary ice production is included via the Hallett-Mossop (rime splintering) process.

In order to investigate the aerosol impacts on the liquid phase alone, the model was modified to have separate categories for aerosol able to act as cloud condensation nuclei (CCN) and ice nucleating particles (INP). Salt was chosen as the aerosol category that would only serve as CCN, as it is totally soluble and cannot act as INP. Dust was chosen as the aerosol acting as INP; routines that allowed liquid nucleation onto dust were deactivated. While dust is known to act as CCN, the DeMott parameterization makes no distinction between immersion and deposition freezing - only the total number of particles, in or out of droplets, is required. Therefore, we think that this separation approach is appropriate. In this study, we are concerned solely with the impacts of CCN on mixed-phase Arctic clouds - this separation of CCN and INP will allow for future study on the impact of INP alone.

In our configuration, radiation is parameterized by BUGSRAD, a two-stream radiation model (Stephens et al., 2001). Subgrid-scale turbulence and diffusion is based on Deardorff (1980) - this scheme parameterizes eddy viscosity as a function of resolved turbulent kinetic energy (TKE). Surface fluxes were set to zero to provide an idealized framework in which cloud processes can be examined without influence from the surface. This is a reasonable approximation for ice surfaces. This removal of surface fluxes also acts to simulate a boundary layer that is decoupled from the surface, which is often seen in the Arctic (Brooks et al., 2017).

The simulations in this study follow a similar setup to those in Sterzinger et al. (2022): a $6 \times 6$ km$^2$ periodic domain with 62.5 m horizontal and 6.25 m vertical grid spacing. Model top was set at 1250 m. The model was initialized with the thermodynamic profile shown in Figure 2a. The cloud layer was added with an adiabatic profile of liquid water over a cloud layer 300 m thick that integrated to a liquid water path (LWP) of 63 g m$^{-2}$; this is similar to the median LWP of 67 g m$^{-2}$ measured over the ASCOS campaign (Mauritsen et al., 2011). Large-scale subsidence prescribed by a fixed divergence rate of $5.0 \times 10^{-6}$ s$^{-1}$.

Simulations were run for a simulated 28 hours with a 1 second integration period. The simulation was initialized to occur on October 1st at 85°N, a location which is in near-total twilight at this time of year. October also corresponds to the time of year when the aerosol concentrations in the Arctic BL are lowest (e.g. Boyer et al., 2023), and as such when entrainment of aerosol from the FT would likely be the most impactful. The model was run for two hours with a quasi-constant aerosol concentration to allow the cloud to spin-up. The prognostic aerosol scheme was turned on after this point, and an additional two hours are given to adjust - analysis in this study is for a 24-hour period beginning at the 4 hour mark.

To test the sensitivity to tropospheric CCN concentrations, a suite of simulations were run across a range of tropospheric salt concentrations. A baseline simulation with a salt aerosol particle concentration of 20 mg$^{-1}$ at all levels was run. Sensitvitity tests were run in which salt concentrations in the FT were set by multiples of 200 mg$^{-1}$ until a concentration of 1000 mg$^{-1}$ (Fig. 2b). These concentrations were chosen to be representative of the range of observed aerosol concentrations in the Arctic troposphere, with 1000 mg$^{-1}$ being a high, but not unrealistically high, value (Figure 1). For all of these CCN sensitivity simulations, dust concentrations were set at 20 mg$^{-1}$ in both the FT and BL.



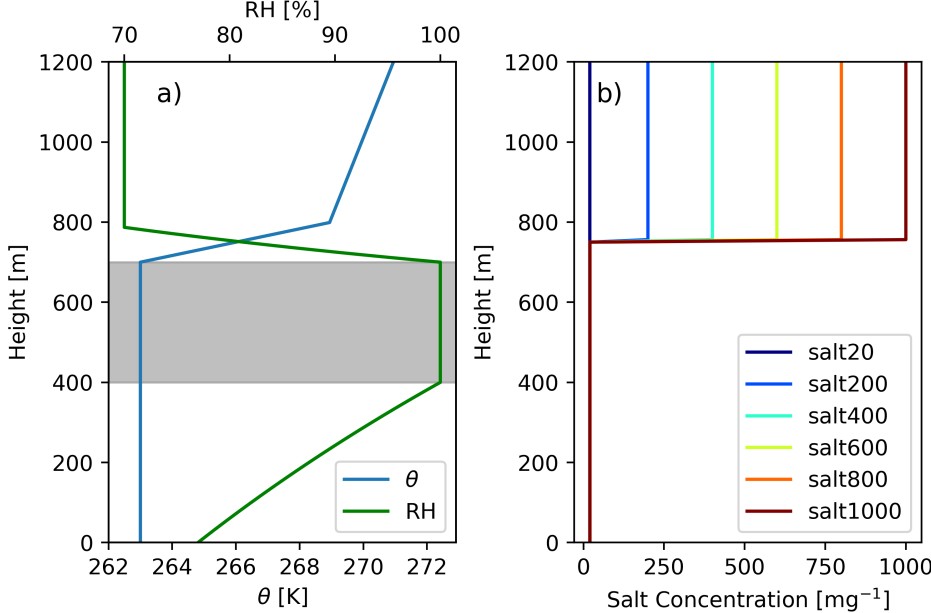

**Figure 2.** (a) Profile of potential temperature ($\theta$, blue line) and relative humidity (RH, green line) used to initialize simulations. Grey indicates the levels initialized as cloudy by adding an adiabatic liquid water profile. (b) Salt aerosol profiles used to initialize each of the simulations.

Since salt concentrations are the only aerosol species being modified in this study, from this point forward any mention of 'aerosol' is in reference to salt/CCN particles alone unless specified otherwise.

# 3 Results and Discussion

## 3.1 Simulation Overview

The clouds produced by the six simulations are shown in Fig. 3. Liquid water mixing ratios (Fig. 3a) are relatively consistent for the higher aerosol concentration simulations, with cloud top mixing ratios of around 0.2 g kg$^{-1}$. While the cloud does display the typical mixed-phase stratocumulus setup of a layer of supercooled liquid above precipitating ice, ice production (Fig. 3b) was quite low and ice masses only reached 0.1 - 0.2 mg kg$^{-1}$. The beginning of the analysis period shows a large amount of ice ($>$0.2 mg kg$^{-1}$), this is residual ice from the quasi-constant aerosol treatment during the spin-up period. After this point, ice production is sustained in salt600 and above, whereas salt20, salt200, and salt400 are unable to sustain substantial ice production.

Figure 4a shows the liquid water path (LWP) of all six simulations from 4-28 hours. There is a strong sensitivity to FT aerosol concentration, with simulations initialized with FT salt concentrations of 20 mg$^{-1}$ and 200 mg$^{-1}$ nearly dissipating within 10-20 hours, while the simulations initialized with concentrations of 400 mg$^{-1}$ or higher are able to persist for the entire simulation period - though salt400 may be headed toward dissipation. The LWP response appears to be non-linear, with



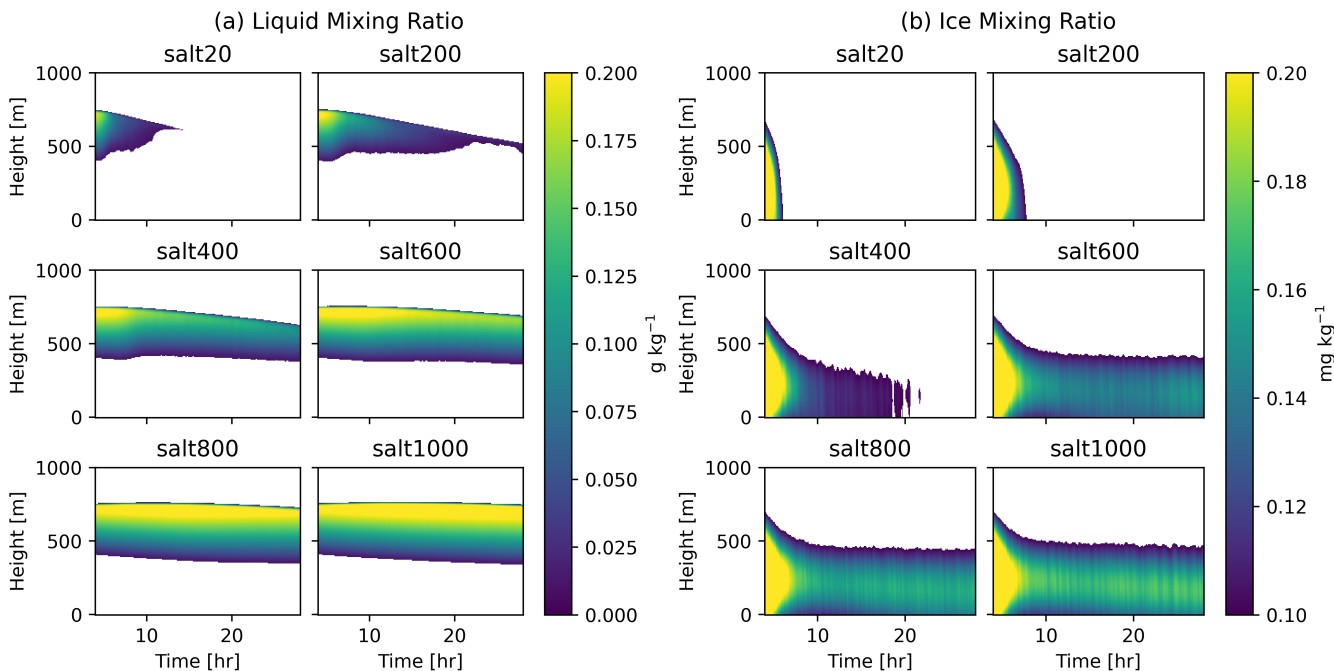

**Figure 3.** Time-height contours of (a) liquid water and (b) ice mass mixing ratio. Regions where cloud water mass is greater than 0.01 g kg$^{-1}$ are considered cloudy, regions where ice water (sum of all ice categories) mass is greater than 0.1 mg kg$^{-1}$ are considered icy. High ice concentrations at the beginning of the analysis period are leftover from high ice generation during the spin-up period.

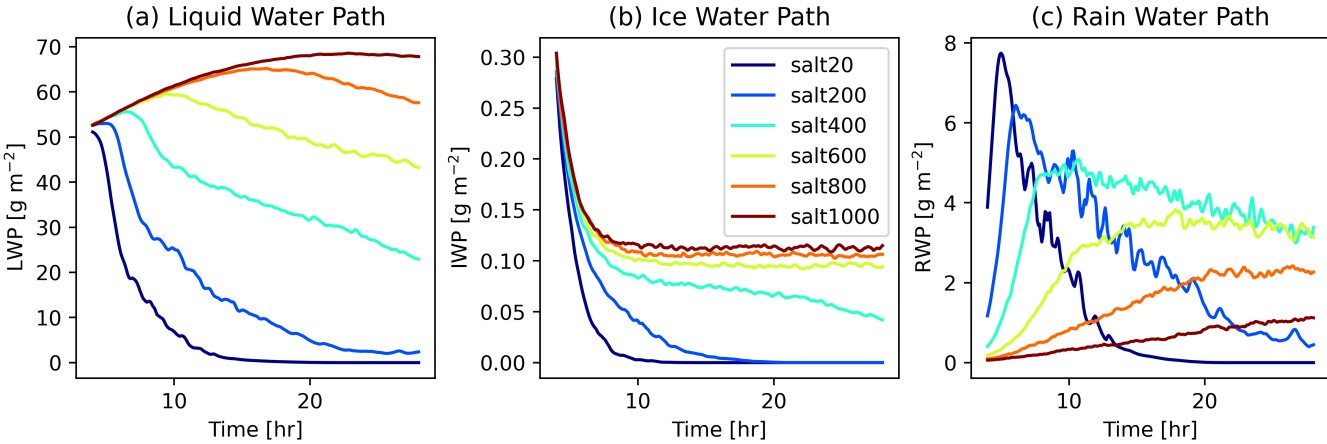

**Figure 4.** (a) Liquid water path, (b) ice water path, and (c) rain water path for each simulation. The spin-up period (first 4 hours) is not shown.





differences between simulations lessening with each subsequent increase in tropospheric aerosol concentration. Salt800 and salt1000 are similar for the first 10-15 hours, but start to diverge after this time. All simulations produce some rain water (Fig. 4c). For salt400 and above, the rain water is about 10% of less of the total liquid water. As seen by a lack of liquid water in the domain mean between the surface and 400m (Fig. 3a), very little rain water actually reaches the surface. Surface precipitation rates are at most 0.25 mm per day, which is essentially negligible and not large enough to be observed. Rather, the vast majority of the little rain that there is is quickly evaporated below cloud base.

Figure 4b shows the evolution of ice water path (IWP) for each simulation. These values are on the extreme low end of typical IWP, a range of 0.1-120 g m$^{-2}$ was reported in Shupe et al. (2008) as the 5th-95th percentile, respectively. Ice number concentrations (not shown) are also low, consistently between 0.1-0.2 L$^{-1}$ after stabilizing from the spin-up period, again about an order of magnitude fewer than typical values around 1 L$^{-1}$, though as the dust concentrations are themselves low (to represent an aerosol-limited environment), this is perhaps not especially concerning. Underproduction of ice is not a new problem - representation of proper ice and liquid quantities together in models has been a persistent issue (Klein et al., 2009; Stevens et al., 2018).

Although dust concentrations were not changed between simulations, there was a response in ice to changes in cloud liquid. As expected, the ice phase of the cloud is dependent on the existence of the liquid phase. Salt20 and salt200, which have rapidly depleting liquid with time (Figs. 3a, 4a) are unable to sustain ice without the liquid water whereas salt600, salt800, and salt1000 all maintain nearly constant IWP after hour 10. Since this study is to investigate the effect of aerosol that act as CCN alone, and such ice mass is very low compared to liquid, the rest of this study will be focused solely on liquid properties and processes.

## 3.2 Surface Aerosol Concentrations

The base simulation, salt20, was initialized with a uniform salt concentration of 20 mg$^{-1}$ in both the BL and FT and dissipated in a manner similar to previously modeled cases of aerosol-limited dissipation (Sterzinger et al., 2022). Since the BL aerosol concentration is initialized to 20 mg$^{-1}$ in all simulations, any changes in cloud liquid properties must come from tropospheric aerosol being entrained into the cloud layer.

Figure 5 shows the domain-mean salt number concentration directly above the surface in the lowest model level (solid lines) as well as the average cloud droplet concentration (dashed lines). Aerosol concentrations decrease in time for the period shown, most likely due to surface deposition and reduction in particle concentrations due to weak collision-coalescence. As is expected, the simulations initialized with higher aerosol concentrations in the free troposphere also have higher concentrations in the boundary layer due to transport of aerosol into the BL via either activation of FT aerosol at cloud top and subsequent hydrometeor evaporation in the boundary layer or by direct transport from the FT without being activated Igel et al. (2017). In all cases the BL aerosol concentration (less than 50 mg$^{-1}$ for all simulations) remains much lower than what was initialized in the FT. There is an approximately linear increase in surface aerosol number concentration similar to the linear increase in initialized FT aerosol shown in Figure 2b.





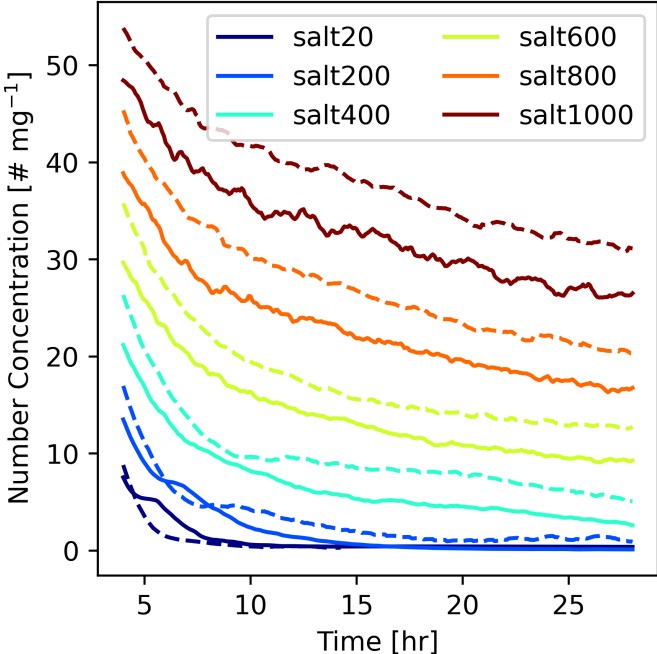

**Figure 5.** Evolution of surface aerosol concentrations (solid) in the lowest model level and mean cloud droplet number concentrations (dashed) within the cloud layer.

These low surface aerosol concentrations show the dependence of a sustained cloud on the above-cloud aerosol. In all simulations except salt 20, the average droplet concentration is higher than the surface aerosol concentration as a result of the entrainment of higher aerosol concentrations at cloud top. Furthermore, we note that a simulation initialized with 20 $mg^{-1}$ of salt in the BL and FT (i.e. salt20) was unable to sustain itself, yet during most of the analysis period salt400 and salt600 - simulations with clouds that persist for the analysis period - have BL concentrations below 20 $mg^{-1}$. An observer with surface data alone could infer that this concentration is what is needed to sustain a cloud, when in reality the cloud is dependent on sustained FT aerosol entrainment to survive.

### 3.3 Precipitation Suppression

The LWP response described in the section above is due largely to a precipitation suppression effect. An increase in aerosol concentrations divides the available water vapor across a larger number of nucleated droplets, decreasing their average size. These smaller, but more numerous, aerosol are less efficient at colliding, coalescing, and growing large enough to fall out as drizzle or rain droplets (Albrecht, 1989). This processes has been observed to occur in warm-phase marine stratocumulus clouds over lower latitudes (e.g. Wood, 2005b), as well as in Arctic mixed-phase clouds (e.g. Peng et al., 2002) - though it's expected that some interactions between different cloud droplet sizes and ice deposition processes make such a process more complex than in liquid-only clouds.



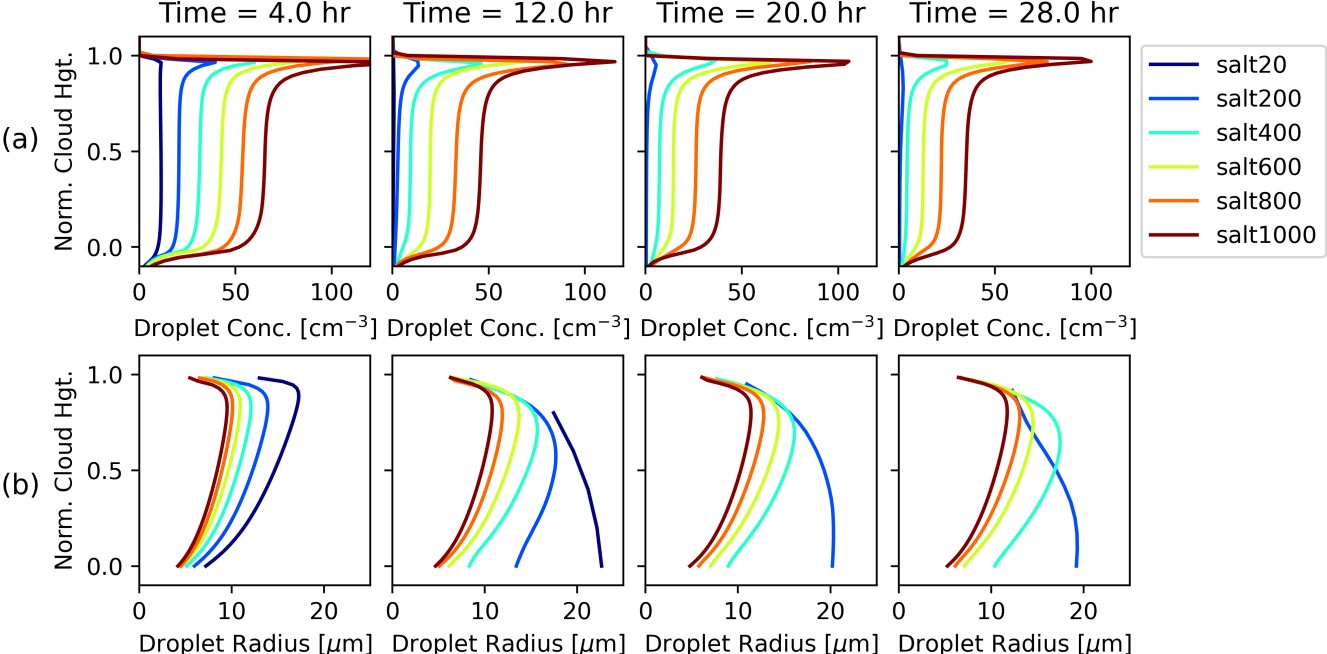

**Figure 6.** Evolution of (a) mean cloud droplet number concentration and (b) mass-mean cloud droplet radius profiles for all simulations every 8 hours. The y-axes display heights normalized to cloud base and cloud top.

Our simulations show an increase in droplet number concentration ($N_d$) and a decrease in mean droplet radius ($r_d$) with an increase in aerosol concentrations. Figure 6a shows profiles of $N_d$ at various times throughout the simulation period, with profiles normalized to cloud top height and cloud bottom. There is an approximate linear increase in the profiles of $N_d$ correlating to the linear increases in tropospheric aerosol. All profiles show a sharp increase in $N_d$ at cloud top, consistent with the nucleation of a relatively high number of entrained aerosol particles. Outside of this layer of enhanced $N_d$, cloud droplet concentrations are relatively constant throughout the cloud, as previously seen in marine stratocumulus (Wood, 2005a). The mean cloud radius (Fig. 6b) decreases with increasing aerosol concentrations. This effect is less pronounced in the simulations with the highest concentrations, as the mean radius scales with $N_d^{-1/3}$ assuming an equal amount of liquid mass being divided between an increasing $N_d$.

In each simulation, cloud number concentrations are decreasing and mean radii are increasing in time. This is indicative of a decrease in the amount of aerosol being entrained into the cloud (this can be seen by the decreasing surface salt concentrations in Fig. 5) and thus a fewer number of cloud droplets being nucleated. This effect is most pronounced in the lower concentration simulations, indicating that there are not enough aerosol being entrained to continue nucleating droplets. Salt200 persists as a very thin cloud, less than 40 m thick and wih LWP $< 5$ g m$^{-2}$ (Figure 4a), with a very small number of relatively large droplets ($< 1$ cm$^{-3}$ in number and 10-20 μm in radius; Fig. 6).





185     The result of the combined increase in cloud droplet number and decrease in radius is a reduction in collision coalescence efficiency. Figure 4c shows the rain water path (RWP) evolution for each simulation. There is clearly sensitivity to the tropospheric aerosol concentration with salt20 and salt200 raining the most at the beginning of the simulation before dissipating, and salt400 and above producing rain throughout. As noted above, although rain is being produced, very little rain actually reaches the surface.

190     This precipitation suppression process is the primary factor in the spread in LWP seen in Figure 4a. With aerosol concentrations too low, cloud droplets become larger and can completely rain out a cloud, such as seen in salt20 and to a lesser extent in salt200. This process is less effective at higher concentrations, as the effects on mean droplet size decreases with increasing number. However, it is not the only process impacting the LWP. Sustained rain production, which is present even in salt1000, does not imply dissipation so long as the rate of rain production is balanced by the production of new cloud water.

195 **3.4   Radiation and Buoyancy**

As is expected, a change in a cloud's amount of liquid water also affects its emissivity. Figure 7a shows a time series of the radiative flux divergence across the cloud layer. This is calculated as the difference in net flux (longwave and shortwave) between cloud top and cloud bottom. The flux divergence is dominated by longwave radiation; as these simulations were initialized at $85°$N in early October, shortwave radiation is nonexistent most of the day and negligible for the 3-4 hours when 200  the sun does peek above the horizon. There is a large spread in the flux divergence, with around 50 W m$^{-2}$ separating salt600 and above from salt20 and salt200 near the end of the simulation. This radiative sensitivity to aerosol concentration is triggered first by the precipitation suppression effect described above. The less numerous, larger droplets created with fewer aerosol lead to the development of thin clouds with less liquid water, which do not behave as a blackbody but rather as a graybody. This radiative behavior of thin water clouds is consistent with previous work (Morrison et al., 2008; Shupe and Intrieri, 2004; 205  Mauritsen et al., 2011; Garrett and Zhao, 2006).

    This longwave sensitivity is important since, in the absence of surface fluxes, the cloud must be maintained from the top-down (cooling at cloud top drives an overturning buoyancy circulation) versus the from bottom-up (surface heat fluxes and BL instability drive vertical motions). As such, the dynamics of the cloud are sensitive to changes in radiative cooling rates within the cloud layer. Figure 8 shows the radiative heating rate profiles every eight hours for the top 20% of the cloud. The 210  higher LWP simulations salt800 and salt1000 show similar radiative heating rates throughout the simulation, but the lower aerosol concentration/LWP simulations produce a range of cooling rates, inline with the fluxes seen in Fig. 7a. Simulations with a decreasing LWP for most of the analysis period (salt20 - salt600) also have a decreasing radiative cooling rate in time. Increased cooling at cloud top drives a stronger overturning buoyancy circulation, further sustaining the cloud.

    The vertical wind variance, which is the vertical component of turbulence kinetic energy (TKE) $\sigma_w^2 = \overline{w'w'}$, thus also 215  has a sensitivity to the tropospheric aerosol concentrations. Figure 7b shows domain-average time series of $\sigma_w^2$. Simulations with higher aerosol concentrations drive stronger average vertical motions. The effect of increasing aerosol concentrations on vertical motions is more apparent at lower aerosol concentrations, where the clouds are thinner and increasing cloud droplet concentration and LWP has a more profound effect on the longwave emissivity of the cloud. As clouds start to approach as





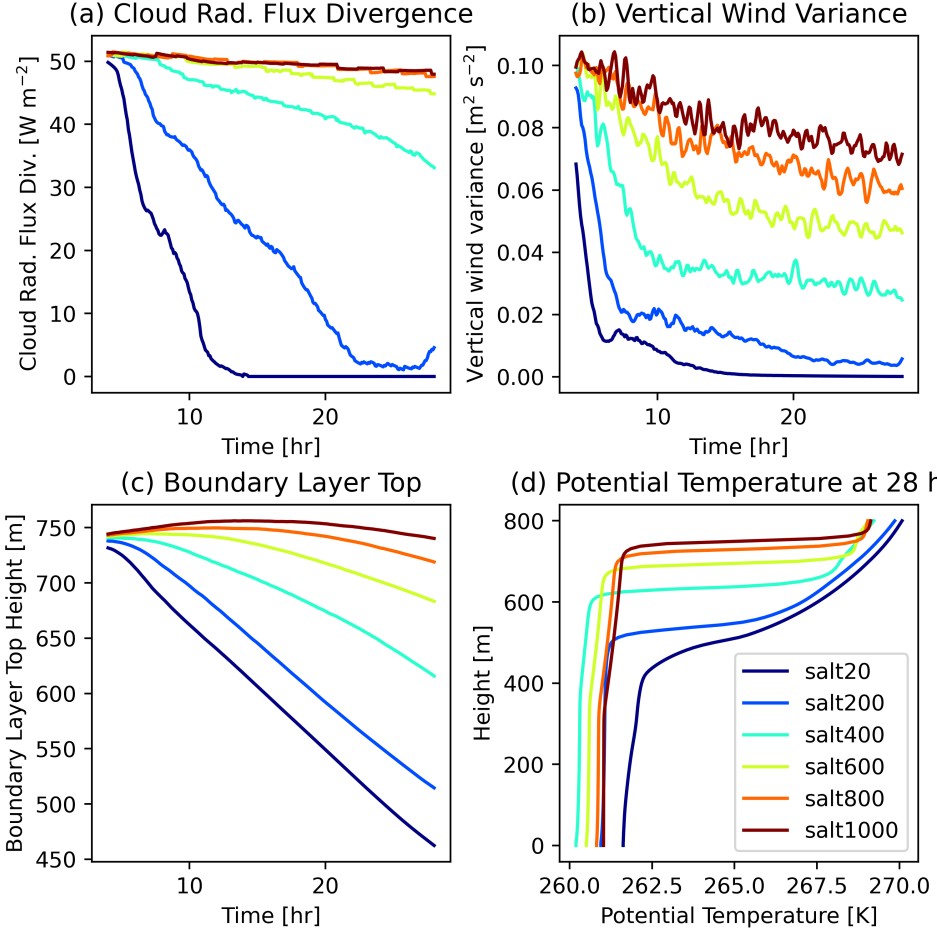

**Figure 7.** Time series of (a) radiative flux divergence across the cloud layer, (b) vertical wind variance ($\sigma_w^2$), and (c) boundary layer top height. (d) Profiles of boundary layer potential temperature ($\theta$) at the end of the simulations.

blackbody in salt600 and above, the difference in $\sigma_w^2$ becomes smaller. Peak $\sigma_w^2$ occurs slightly below cloud top (not shown);

at this location the negatively-buoyant downdrafts from cooling near cloud-top are at their strongest and these downdrafts drive the boundary layer circulations.

These radiative and dynamic effects are not responsible for the initial LWP response to aerosol, but are instead created initially by the precipitation production suppression and act as a positive feedback. This leads to stronger vertical motions in simulations with higher tropospheric aerosol concentrations. These stronger vertical motions help drive additional cloud

droplet condensation, increasing LWP and feeding back into the cycle.

While generally longwave impacts of the aerosol indirect effects are seen as minimal (especially in thicker stratocumulus clouds in lower latitude), Morrison et al. (2008) found through modeling that changing aerosol concentrations had a longwave effect in thin clouds with LWP $< 50$ g m$^{-2}$. Shupe and Intrieri (2004) have a lower threshold of 30 g m$^{-2}$ for this effect. Our



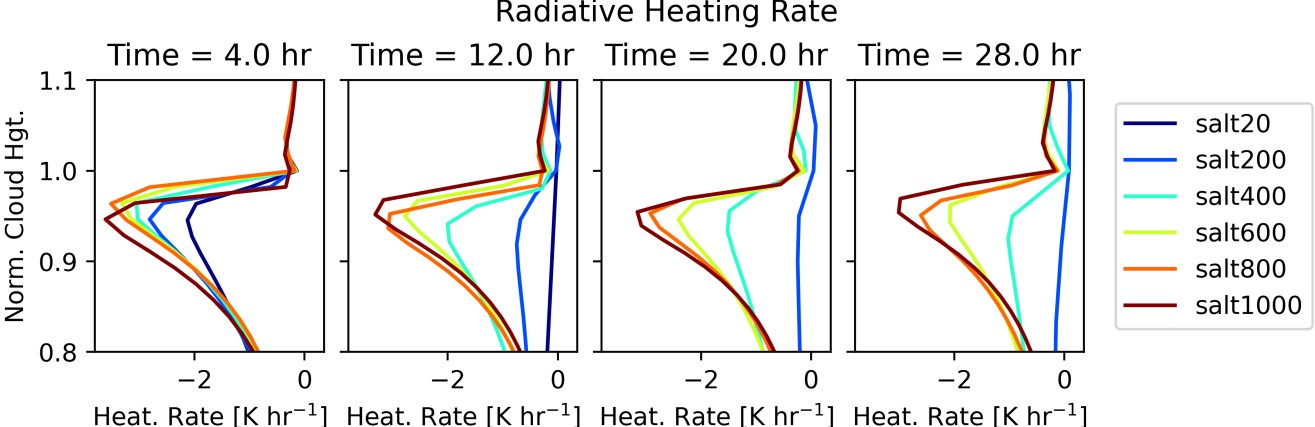

**Figure 8.** Radiative heating rate profiles every 8 hours of simulation time in the top 20% of the cloud layer. The heights are normalized to cloud top and bottom.

results are consistent with these previous studies. Salt400, with its LWP of 30-50 g m$^{-2}$ throughout most of the simulation, has

a flux divergence that differs substantially from those for salt600 and above. Mauritsen et al. (2011) also found that for clouds with low CCN concentrations, a tenuous cloud regime exists where radiative forcings start to tend towards zero faster than expected from the aerosol-cloud-albedo and cloud-lifetime affects alone. Garrett and Zhao (2006) also found that thin Arctic cloud emissivity is sensitive to changes in aerosol concentrations, as cloud emissivity ($\varepsilon$) changes with droplet size and as such these clouds act as a graybodies ($\varepsilon < 1$) not blackbodies ($\varepsilon \approx 1$) at small drop sizes.

### 3.5 Boundary Layer Structure

The combined effects of increasing FT aerosol concentrations changes the boundary layer structure between simulations. Figure 7c shows the evolution of boundary layer top height (defined as the height of the base of the temperature inversion) with time. In the higher aerosol simulations (salt800/salt1000), this is nearly constant in time. On the other extreme, salt20's boundary layer top decreases nearly 300 meters in a 24 hour period, or around 0.35 cm/s. Accompanying the lowered BL tops

are weakened temperature inversions (Fig. 7d). These chagnes are unsurprising given the other results so far. With surface fluxes disabled and without a cloud-driven circulation to drive entrainment, the large-scale subsidence will act to lower the height of the inversion and a lack of mixing will weaken the inversion. Combined, these changes suggest that the boundary layer is collapsing in salt20.

The collapsing of the boundary layer has implications for aerosol entrainment. Successful entrainment of tropospheric

aerosol depends on a layer of enhanced aerosol concentration directly above the cloud. Figure 9 shows that in salt20, salt200, and (to a lesser extent) salt400 a buffer develops between the aerosol in the FT and the top of the boundary layer. In these simulations, the boundary layer is collapsing faster than the layer above can be replenished with aerosol by subsidence. In salt600 and above, the boundary layer top is not lowering as quickly and is better able to stay in contact with the tropospheric





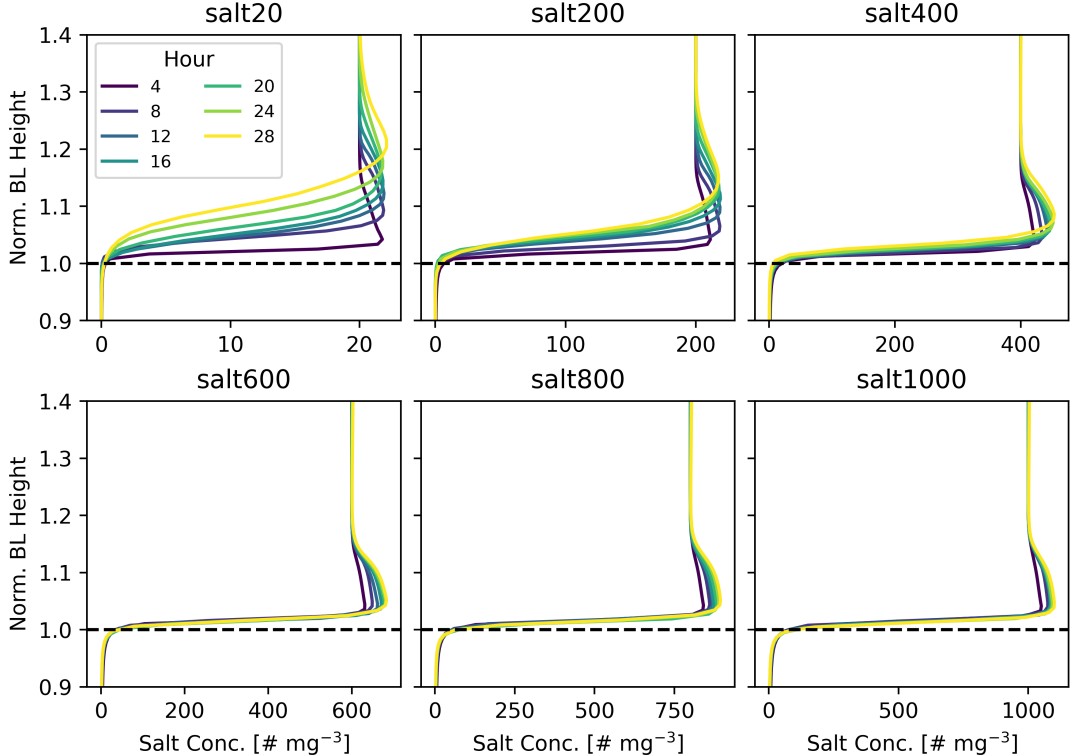

**Figure 9.** Profiles of aerosol concentrations near the boundary layer top. Heights are normalized with respect to the boundary layer top.

aerosol reservoir. This is a likely factor in the faster decrease of BL aerosol concentrations in the salt20 and salt400 simulations
seen in Figure 5, and, more importantly, in the ability of clouds to sustain themselves in the face of very low boundary layer
aerosol concentrations.

## 4 Conclusions

We present idealized LES simulations of an Arctic mixed-phase cloud with various tropospheric salt concentrations (which
we refer to generally as 'aerosol', and which serve only as CCN). A baseline simulation with low aerosol concentration (20
mg$^{-1}$ of salt) in both the boundary layer and free troposphere simulated a cloud that was unable to sustain itself more than a
few hours. Increasing tropospheric salt concentrations from 200 - 1000 mg$^{-1}$ (in multiples of 200 mg$^{-1}$) resulted in a positive
LWP sensitivity. The lower aerosol concentration simulations (200 mg$^{-1}$ and below) yielded clouds that either did dissipate
within the simulation period or were declining enough with respect to LWP, IWP, and radiative cooling rates that they were not
likely to persist for much longer had the simulations been run for additional time. The higher aerosol concentration simulations
(600 mg$^{-1}$ and above) produced clouds that had more stable LWP and IWP. We find that tropospheric aerosol concentrations
of more than 200 mg$^{-1}$ were necessary for cloud persistence beyond about 24 hours. These concentrations are well within the




range generally found in the lower free troposphere (Figure 1b and e.g. Lonardi et al., 2022; Jung et al., 2018). Given that the required concentrations are realistic, aerosol entrainment from the FT is likely important in the summertime high Arctic for maintaining low-level clouds.

The cloud sensitivity to aerosol in the free troposphere is a result of entrainment and activation of aerosol particles from the troposphere into the cloud layer. This process causes three feedbacks that result in the change in liquid water content in the cloud:

- Increasing tropospheric aerosol concentrations leads to the commonly noted precipitation suppression effect. As more aerosol are entrained into the cloud layer and activated, the available liquid is divided among more droplets, causing
an increase in cloud droplet number and a decrease in their size. This results in a less efficient collision-coalescence processes and thus less removal of water by rain.

- As a consequence of the rain suppression the higher liquid water content in the higher aerosol concentration simulations causes stronger cooling at cloud top. This cooling, which is primarily responsible for the circulations that maintain the cloud in the absence of surface forcing, drives stronger vertical motions in the clouds with higher droplet concentrations.
This processes is kickstarted by the change in liquid content caused by the precipitation suppression described above.

- Finally, due to these two processes, higher FT aerosol concentration simulations are better able to maintain contact between the FT aerosol reservoir and the boundary layer top in order to maintain the very aerosol entrainment that causes the precipitation suppression.

There is potential for tropospheric aerosol to be even a larger source of CCN due to the formation of a "cloud inside
inversion" (CII) regime, which is a dominant regime over Arctic sea ice (Sedlar et al., 2012) but which was not simulated here. The presence of the cloud layer inside the inversion means that cloud droplets could nucleate directly within the free troposphere, without needing to be entrained through a temperature inversion first.

Our simulations produced surface aerosol concentrations that were representative of cloud droplet number concentrations. This is due to the coupled nature of the simulated boundary layer; had it been decoupled, the enhanced aerosol concentrations
making their way into the boundary layer would likely not have been detectable at the surface. The observed surface concentrations were still much lower than those of the FT. In a real world situation similar to the one simulated, surface measurement of aerosol concentration - even alongside cloud droplet number concentration - would indicate that the cloud could persist with aerosol concentrations of 20-50 mg$^{-1}$. We find that simulations initialized with concentrations that low throughout the atmosphere do not produce stable clouds. Surface based measurements of this sort would not necessarily suggest the influence
of FT aerosol on the cloud, despite it being necessary for stable cloud persistence. Even the most extreme simulation with initial FT aerosol concentrations of 1000 mg$^{-1}$ (which has been observed to occur in the Arctic in Fig. 1b), producing the most stable cloud, yielded surface aerosol and cloud droplet number concentrations $< 50$ mg$^{-1}$.

We also note that the low ice mass and concentrations in this study simplified the discussion of processes impacted by increases in tropospheric aerosol. The liquid response would have likely been complicated in the presence of more ice, where

processes such as the Wegener-Bergeron-Findeisen (WBF) process - in which cloud ice grows at the expense of evaporating liquid droplets - would likely have played a larger role in the cloud response.

  The separation of aerosol in our methodology between those that can act as CCN and INP allows for a natural continuation of this study in which salt (CCN) concentrations and dust (INP) concentrations are varied in the free troposphere. In reality, of course, there is no such strict partitioning between CCN and INP, and the measurements of increased aerosol concentrations
above the Arctic boundary layer naturally include both aerosol that can act as CCN and INP. Running similar modeling studies with more realistic aerosol treatment is critical to fully understanding the impact of these aerosol on Arctic cloud properties.

*Code and data availability.* Model source code and namelists used in this study can be found at https://doi.org/10.5281/zenodo.7991355 (Sterzinger et al., 2023). Horizontally-averaged processed model data used for analysis can be found at https://doi.org/10.5281/zenodo. 7996451 (Sterzinger and Igel, 2023). Code used to generate plots for this paper can be found at https://doi.org/10.5281/zenodo.7996595
(Sterzinger, 2023).

*Author contributions.* LS and ALI conceived the study. LS conducted and analyzed the simulations. ALI analyzed the tethered balloon data. LS wrote the original draft. LS and ALI edited and reviewed the draft.

*Competing interests.* The authors declare no competing interests.

*Acknowledgements.* This research was supported by the U.S. Department of Energy's Atmospheric System Research, an Office of Science
Biological and Environmental Research program, under Grant No. DE-SC0019073-0.



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
