# Peer review of "Above Cloud CCN Concentrations Help to Sustain Some Arctic Low-Level Clouds"

_EGUsphere, 2023_

## Author Comment (AC1)

This paper describes a hypothetical numerical Large Eddy Simulation, to test the hypothesis that Arctic clouds are sensitive to entrainment of free troposphere aerosols from remote sources. There is nothing fundamentally wrong with the paper; I just can't help wonder why this study was perceived, why now, and what new we can learn from it? There are a lot of detailed results that seems a bit in lack of a plan how to use, and I think the study needs to develop before it can be published; hence "major revision".

Thank you for the critical comments. They point to a need to better motivate the study and frame the conclusions. In response, we have substantially modified these parts of the paper.

Additionally, in response to comments from both reviewers, we decided to make major changes to our simulations. The surface temperature has been increased to 273.15K, a value that is frequently measured in the transition seasons over ice in the Arctic. Solar radiation is now turned off, as suggested by this reviewer, in order to simplify the setup and remove the need to specify a date associated with the simulations. Ice processes remain on, but because these simulated clouds are only slightly supercooled, the ice is unimportant and is now completely neglected in the presentation of the results. A small suite of sensitivity tests is presented in which the initial thermodynamics conditions and the subsidence rate are modified. Among these are simulations that are very similar to our original simulations that used a surface temperature of 263 K. We do not intend for these sensitivity simulations to be a complete investigation of the sensitivity of our results to these choices, but they do serve to illustrate possible alternative outcomes.

Main concerns

The study essentially falls a part into three stories: 1) one that deals with the entrainment of aerosols from the free troposphere into the boundary layer; 2) a second that deals with the effects of those aerosols on the liquid water clouds in this boundary layer, and; 3) third, the results, as a consequence of 1 & 2, on the boundary-layer structure.

For all parts, one problem is the case and the LES. There are no attempts to illustrate or ascertain that this LES in this setup is capable to simulate this cloud and that the processes that makes it tick are adequatly modeled. The case itself looks a lot like a midlatitude marine stratocumulus but is placed at a specific date and specific location in the Arctic. But there is no information on how the case was designed and why these choices were made. The results are dependent on the rate of entrainment, which is dependent on the model formulations, the setup (e.g. subsidence) and the resolution

etc. If the entrainment is sensitive to for example resolution, then all the results here will also be.

We have added more justification for the set up in the manuscript. Our setup is based on our experience with the real simulated cases in Sterzinger et al 2022 in which the shallowest boundary layer was about 600m, all were well-mixed or well-mixed with a shallow stable layer at the surface, and liquid water paths were ~25-100 g/m$^2$. Many of our choices are also supported by Jozef et al. 2023. Please see the revised manuscript for more details. The date and place are unimportant except insofar as it sets the solar radiation. We agree that there are similarities to midlatitude marine stratocumulus setups, but we note that in both the original simulations and the new simulations, we included a moisture inversion above cloud top which is a feature that is typically only found in the Arctic. That inversion was not obvious in the original submission since we only showed the initial relative humidity profile.

In response to this comment, we have added a small suite of sensitivity tests which include modifications to the subsidence rate and thermodynamic conditions. These include tests in which the temperature inversion strength is halved, the surface temperature is decreased by 10K, and a stable layer is introduced below the cloud layer.

And what is new here? The second author already a few years ago showed, using another LES but on a well-documented real observed case, that free troposphere aerosols are indeed entrained into the PBL where they have an effect on the clouds. In that paper, as also in this, one main conclusion was that aerosol observations near the surface does not necessarily provide any guidance as to what goes on in the cloud layer, especially if this is dynamically decoupled. So what does this paper add that wasn't shown already in the previous paper? And why should aerosols not be entrained? At the interface of a turbulent boundary layer to a laminar free troposphere there is always entrainment. How much may depend on how vigorous the turbulence is and how well resolved the stably stratified interface is resolved.

Yes, Igel et al (2015), the second author's paper, already showed that the aerosols can be entrained. However, it is incorrect to state that this paper also looked at the impact on the cloud properties. It did not and in that respect, this paper adds to the previous study.

We agree – it is not surprising that the aerosols are entrained. As the reviewer says, why wouldn't they be entrained? The question though is not will they be entrained, but rather, will they be entrained quickly enough (when present at realistic concentrations) to impact the lifetime of the cloud? Is their presence necessary or sufficient for the

maintenance of the cloud? These questions are now more clearly made in the revision. Our sensitivity tests allow us to see how the results change with different thermodynamic and turbulent conditions.

So, what is new here? Sure, there are more runs here and more details but how are those used to shed more light on the delicate balances at the cloud top? It looks to me that the synoptic scale divergence prescribed has a large effect on the result; maybe there should have been less runs with different aerosol settings and more runs with different subsidence? And why was not inversion strength varied or the cloud bulk features(?); none of the simulated clouds are very dense. So, the results show that entrainment of free troposphere aerosols happens (provided there is any right at the interface), but any boundary-layer meteorologist could have told you that! What else did we learn?

Again, the point was not to learn that the entrainment happens, please see the above responses.

Once inside the boundary layer these aerosols have more or less the expected effects. Ice is there but IN is so small that ice plays no important role to the dynamics here. So why have it there? Solar radiation is also present, but so little it (probably) do not have an effect. So why have it there at all? More free troposphere aerosol results in more CCN in the boundary layer – obviously – and this results in denser clouds (larger LWP) with smaller and more numerous droplets. And this in turn leads to more cloud top buoyancy and more mixing; Nothing spectacular here; mostly the expected effects. But lots of details that may – or may not – reveal something. Problem is those are never explored in any detail; the analysis is unimaginative and the result is rather boring. Probably right, at least in principle, but still rather boring. With this effort in numerical simulation, there just has to be something more you can do.

We agree, the ice plays a minimal role and we also agree that the solar radiation plays a minimal role. We have taken the reviewer's suggestion to turn off solar radiation but choose to keep ice processes.

Minor comments:

Line 8: That the cloud top does not move is controlled by subsidence. In reality this mean that the cloud does move upward by mixing, but that i is immediately advected back down again by synoptic-scale advection. The bulk result may be close to net zero, but mixing and advection are not the same thing.

We have removed "does not lower rapidly in time such that it" from the abstract.

Line 15: Clouds are indeed important for Arctic climate but I have seen no study that shows them to have a large effect on "Arctic amplification". If these authors have, I'd like to see a reference here.

We have removed these statements as part of our rewrite of the introduction.

Lines 20-21: A reference should be inserted here, as this statement comes from one single study (SHEBA) and most other summer studies I have seen does not show this warming regime, unless the ice has melted completely.

We have removed these statements as part of our rewrite of the introduction.

Line 45: "argued" is better than "found"; there is no way in which Matt could have proven this, but it is a good argument.

Agreed and changed.

Line 64: What "scale" is that? LES is a numerical technique and is appropriate for flows with large enough eddies but there is no specific scale; it all depends on the problem.

This statement has been removed.

Lines 73-74: So what happens with coalescence? Each droplet forms on one CCN each, but after having collided, forming a larger drop, there is no way the new aerosol formed by evaporation of that drop can be assigned back to the original CCNs.

RAMS tracks the aerosol mass contained within hydrometeors such that aerosol mass is contained within rain. When the rain evaporates, the particle returned to the atmosphere is should be larger than the particles that were originally activated to form the droplets that then collided to form the raindrop.

Line 69: I'd like to know more about the aerosols scheme. How is formation of new aerosols handled and how are aerosols that become CCN or are entrained replaced? How is size distribution consndered?

We have added more information regarding the aerosol scheme. In short, we do not have new particle formation – it is clearly a limitation of the study. The size distribution is assumed to be lognormal. The mean size changes in time, but the distribution width is a constant. More information has been added to the manuscript.

Line 81-83: This is a different motivation to that given earlier, and does not rely on assumptions on entrainment.

We're unsure what the reviewer's concern is here.

Line 84: Is this radiation code sensitive to effective droplet radius?

Yes. This is now explicitly stated.

Line 86-87: Setting the heat fluxes to zero is a reasonable assumtions; they are never very large anyway. But what about the momentum flux? Is this a free-slip simulation? How is this "reasonable" for ice surfaces?

The roughness length for momentum is set to 0.0005m based on Schroeder et al 2003 (https://doi.org/10.1029/2002JC001385). This same study finds that the roughness lengths for sensible and latent heat fluxes are 3-4 orders of magnitude lower, which justifies our choice to neglect these fluxes (as the reviewer points out).

Line 94: It looks like the cloud top is decreasing, at least for the less vigorous boundary layers. So, how was this value selected and how is the LES upper boundary and the magnitude of the entrainment affected? Is it a special target to have a constant cloud top constant, and should that then not require different divergence? And how would that affect entrainment?

The divergence rate was selected somewhat arbitrarily, our main concern being not to choose a value so low that the cloud top hit the model top. We now included sensitivity tests to the value that we chose.

Line 102: In the whole intro with references and all, the units for aerosol concentreations is $cm^{-3}$. Here all of a sudden, in the LES, it is $mg^{-1}$. Why?

RAMS takes number concentration input in $mg^{-1}$, but it is typical to discuss concentration in $cm^{-3}$, so we use that unit elsewhere.

Line 105-106: Since IN and ice processes are hardly considered, why even bother with IN? This is hardly a prototypical mixed-phase cloud as it is.

We have removed the discussion of ice processes.

Line 156: Delete space between "salt" and "20".

Thanks, this has been done.

Line 165: Do you mean "dropets" and not "aerosols" here?

Yes, thank you.

Line 176: What "cloud radius"?

We mean "cloud droplet radius". This is corrected.

Line 179: What "cloud number concentration"?

Similarly, we mean "cloud droplet number concentration". This is corrected.

Lines 198-199: Why choose such an awkward date? While it is true that aerosol concentrations are low in autumn, this is a hypothetical LES and it would not be more or less hypothetical with the sun switched off completely.

No date is used in the revision.

Line 200: The flux divergence cannot have the unit "W m$^{-2}$".
True, the unit is typically W m$^{-2}$ m$^{-1}$. Here we have only taken the difference in flux between cloud top and cloud base, but not divided by the cloud thickness, so the unit is accurately reported. We applied the term "flux divergence" despite not dividing by the thickness as has been done by others, e.g. Zheng et al 2021 (https://doi.org/10.1029/2021GL094676). We have changed the term to "flux difference." Note that this quantity is equivalent to the integrated cooling rate.

---

## Author Comment (AC2)

Thank you to the reviewer for these comments. In response to comments from both reviewers, we decided to make major changes to our simulations. The surface temperature has been increased to 273.15K, a value that is frequently measured in the transition seasons over ice in the Arctic. Solar radiation is now turned off, as suggested by Reviewer #2, in order to simplify the setup and remove the need to specify a date associated with the simulations. Ice processes remain on, but because these simulated clouds are only slightly supercooled, the ice is unimportant and is now completely neglected in the presentation of the results. A small suite of sensitivity tests is presented in which the initial thermodynamics conditions and the subsidence rate are modified. Among these are simulations that are very similar to our original simulations that used a surface temperature of 263 K. We do not intend for these sensitivity simulations to be a complete investigation of the sensitivity of our results to these choices, but they do serve to illustrate possible alternative outcomes.

This study uses LES to simulate idealized mixed-phase clouds in a relatively clean Arctic boundary layer environment. Sensitivity tests are conducted to explore the impact of higher tropospheric aerosol concentrations to cloud properties. Aerosols in the boundary layer is kept 20 per mg while it is increased up to 50x in the free troposphere. Results show that entrained aerosol from the free troposphere can suppress precipitation and help to sustain clouds for longer time. I think the simulation results are clear and make sense to me. My major concern is how those simulations are relevant to real clouds in the atmosphere.

I'm skeptical about the relatively high droplet number concentration at the cloud top from the simulations (Figure 6). Are those simulations realistic? Therefore, I read some MOSAiC-related papers cited in the manuscript to learn more about the observations. I might miss some other papers, but I do not find observational papers to support model setups and conclusions in this study. I recommend the authors to add more observations to justify their model setup and/or conclusions. Some specific comments are listed below:

1. Figure 1 is the only figure related to the observation. I have several questions: do all those days have clouds? Where is the cloud layer on each day? What about the profile of large aerosol particles (those can contribute to droplet formation), instead of all aerosols larger than 12 nm? I recommend the authors plot similar figures like Figures 6, 13, 14 in Lonardi et al. (2022), for all cases chosen in this study. The model setup and conclusions would be more convincing if the authors can show observational evidence of the existence of aerosol layers above the cloud layer, and/or the potential impact of aerosols in the free troposphere to clouds in the boundary layer.

Yes, as best we can tell, all of these days have clouds, but we don't believe that all of the data was taken in cloudy conditions – we suspect that in some cases the data may have been taken during a clear period. Regardless, we have updated Figure 1 to include the N12 and N150 concentrations as well as an estimate of the location of cloudy layers. Estimating the cloudy layers was difficult to do for most flights because in situ cloud data are typically not measured. Estimates come primarily from broadband longwave fluxes, in some cases in combination with RH data or the VIPS cloud flag (only available for two flights analyzed here). The broadband fluxes and VIPS data are typically not coincident with the aerosol data but rather typically come from 1-3 hours earlier since not all instrumentation could be flown together on the tethered balloons.

We have moved the discussion of these observations to a new section (#2) and provided more information about the data sources and more discussion of the data as it relates to this study. While we have not exactly reproduced Figures 6, 13, and 14 of Lonardi, we have added new panels d-f to briefly analyze the size distribution of the particles with respect to the mixed layer tops. Here is the revised Figure 1:

[Figure]

Figure 1. Profiles of (a) potential temperature, (b) concentration of particles with diameter >12 nm (N12), (c) concentration of particles with diameter >150 nm (N150), and (f) the ratio of N150 to N12 for select tethered balloon flights during the MOSAiC campaign. Black outlined circles represent the top of the mixed layer for each profile. Thin black lines indicate the most likely location of a cloud layer. (d) Normalized distributions averaged over 100m above the mixed layer top and (e) the normalized size distribution averaged over 100 m above the mixed layer top minus the normalized size distribution averaged 100 m below the mixed layer top.

2. Based on Lonardi et al. (2022) Figure 6, clouds on July 23 and July 24 are all above the top of the boundary layer. Therefore, I think Figure 1 might be misinterpreted by the readers that clouds are in the boundary layer and they are affected by aerosols above during the MOSAiC campaign.

   Yes, the clouds are in a mixed layer that is decoupled from the surface. We have clarified the wording and now explicitly included an estimate of where the clouds are in Figure 1.

3. The authors said that "Here we extend the analysis presented by Lonardi et al. (2022)..." However, based on Lonardi et al. (2022) Figure 6, the large temperature inversion on July 23 and July 24 are at about 600 m and 900 m, respectively. However, in Figure 1 of this study, the top of the boundary layer on these two days are at about 300 m and 100 m. Please explain why they are so different.

   For July 24, this was our error – we had correctly identified the large temperature inversion, but that height was modified in a transcription error that we did not catch. It is corrected now. For July 23, the difference arises because there are two datasets for July 23, and we used the second dataset, not the first which is shown in Lonardi et al. We had unconsciously excluded the first because the meteorology data file is formatted differently which led to a problematic read of the data, but one that did not throw errors in our script, only produced NaNs. This flight is now included. We do note though that our estimate of the cloud layer is different from that shown in Lonardi et al 2022. The cloud layers are our own best estimate given that the Lonardi et al. 2022 methods are not documented.

4. Figure 2. Are the initial profiles of potential temperature and relative humidity based on the observation durign the MOSAiC campaign? Based on Lonardi et al. (2022) and also Figure 1 in this paper, surface temperature is very close to 0 C and the cloud temperature is just slightly below 0 C. But the initial temerapture profile for simulations in this study is much lower. Please justify the model setup.

   No, the profiles are not based on the data from Lonardi et al. (2022). Rather, we used idealized profiles for initialization. These profiles have been modified from the originals in the revised submission. More justification is provided in the main text.

5. Please provide formules of the profiles such that simulations can be rerun by others. What about the initial wind profiles? Do you nudge those profiles?

$$\theta(z) = \begin{cases} \theta_0, & z \leq 700\text{m} \\ \theta_0 + a(z - 700), & 700\text{m} < z \leq 800\text{m} \\ \theta_0 + 100a + 0.005(z - 800), & z > 800\text{m} \end{cases}$$

$$w(z) = \begin{cases} w_0, & z \leq 700m \\ w_0 + \dfrac{0.75w_s(800) - w_0}{100}(z - 700), & 700\text{m} < z \leq 800\text{m} \\ \dfrac{0.75}{2}w_s(z)\left(\exp\left(-\dfrac{z - 800}{200}\right) + 1\right), & z > 800\text{m} \end{cases}$$

In the base set of simulations, $\theta_0$ = 273.15K, a = 0.06 K/m, and $w_0$ is set by the mixing ratio that gives 100% relative humidity at cloud base. The excess water vapor in the cloud layer is converted to liquid water by RAMS at the associated latent heat is added to the temperature profile. This is perhaps an unusual way to initialize with a cloud layer, but the model succeeds in producing typical looking boundary layer profiles quickly. In the reduced temperature simulations, $\theta_0$ = 263.15K. In the weakened inversion simulations, a = 0.03 K/m. In the stable layer simulations, we modify the equations for the boundary layer:

$$\theta(z) = \begin{cases} \theta_0 + 0.015z, & z \leq 500\text{m} \\ \theta_0 + 7.5, & 500 < z \leq 700 \\ \theta_0 + 7.5 + a(z - 700), & 700\text{m} < z \leq 800\text{m} \\ \theta_0 + 7.5 + 100a + 0.005(z - 800), & z > 800\text{m} \end{cases}$$

$$w(z) = \begin{cases} w_0 - 4\text{x}10^{-6}(700 - z), & z \leq 500m \\ w_0, & 500\text{m} < z \leq 700\text{m} \\ w_0 + \dfrac{0.75w_s(800) - w_0}{100}(z - 700), & 700\text{m} < z \leq 800\text{m} \\ \dfrac{0.75}{2}w_s(z)\left(\exp\left(-\dfrac{z - 800}{200}\right) + 1\right), & z > 800\text{m} \end{cases}$$

where the parameters are defined in the same way as for the base simulations. The water vapor gradient in the stable layer was chosen to give a relative humidity profile that is similar to that of the other simulations.

Winds are set to zero. None of the profiles are nudged.

All of the profiles used to initialize the RAMS simulations are specified in files beginning with "SOUND_IN". These files are located in the code repository at https://doi.org/10.5281/zenodo.7991354.

6. Figure 6. Is there any observational evidence to show that cloud droplet number concentration is maximum near the cloud top?

We are not aware of any observations of coincident aerosol and droplet concentration measurements in high Arctic clouds which could be used to compare with our simulations. However, for reasons that we did not investigate, the enhancement in droplet concentration near cloud top is much reduced in our new simulations, and we suspect that it would be difficult to detect in aircraft-based observations.

7. Page 4, Line 102, what is the CCN size distribution in the model? Is it fitted based on observations? I think results are sensitive to the CCN distribution. If the authors do not test its sensitivity, it should be clearly stated.

We do not test the sensitivity to the CCN distribution or composition. The sea salt aerosol is lognormally distributed with a modal diameter of 200 nm and a standard deviation of 1.5.

8. Entrainment rate is critical to bring aerosols from the free troposphere to the boundary layer. It would be nice to plot the time series of entrainment rate for different cases.

We at one point did have a time series of the entrainment rate, but felt that it didn't add anything beyond what is shown with the BL top time series. The entrainment rate can be inferred from this time series.

9. Page 9, Line 180: "This is indicative of a decrease in the amount of aerosol being entrained into the cloud…" Do you know the amount of aerosols entrained from the free troposphere as a function of time?

No, we do not know the aerosol entrainment rate. We have modified this statement to read "In each simulation, cloud number concentrations are decreasing and mean radii are increasing in time. This is indicative of either a decrease in the availability of CCN in the boundary layer and/or a decrease in the amount of aerosol being entrained into the cloud."

10. Page 10, Line 215: "Figure 7b shows domain-average time series of $\sigma_w^2$". People usually calculate $\sigma_w$ in the boundary layer where turbulence is vigorous. Just want to make sure $\sigma_w$ is averaged in the whole domain or just in the boundary layer? It makes more sense to only average in the boundary layer. It would be better to show the $\sigma_w$ profile.

We've taken your suggestion and replaced the time series with the average vertical profile at the end of the simulations.

Other comments:

1) Page 7, Line 127: delete "is"

   The sentence was grammatically correct with the double 'is' but was certainly awkward. We have rephrased the sentence.

2) Page 7, Line 130; Page 11 Line 219: "not shown" is not acceptable. Please consider adding it in the main text or supplement.

   With the new simulations, ice is even less important than before. We have removed this statement, but note that we no longer show any results regarding ice.

3) Page 12, Line 230: "Mauritsen et al. (2011) also found that..." This sentence is not clear to me. Please check.

   This sentence has been removed.

4) Page 12, Line 246: "In these simulations, the boundary layer is ..." This sentence is not clear to me. Please check.

   The sentence now reads: "In these simulations, the boundary layer is collapsing which leaves behind a layer of air with aerosol concentrations that are much lower than those is the rest of the free troposphere."

---

## Author Response (AR2)

This is the second time I read this, and very I'm happy to say it reads much better this time. After my first read i felt the changes still needed could be easily done by editing and I was leaning towards "minor revision". However, summarizing my finding they now appear somewhat more substantial and there are issues that must be resolved. Hence, I'm going for "major revision" but will let the editor know I would survive if the decision came down to be "minor".

Major comments:
While my first reaction to a summertime case without solar radiation was quite negative, and I don't thing I explicitly suggested that, I must say as long as its clear that this is a hypothetical experiment, is is quite successful.

However, the sensitivity case with a stable near-surface PBL is quite unrealistic; I've seen many profiles of clouds like this and I've never seen one like it. I can't see what process that could lead to this shape of profiles, and it does have the unfortunate consequence of making a much juicier cloud than all the other cases. I instead suggest lowering the surface temperature by that amount, rather than increasing the cloud temperature. That would serve to preserve - roughly - the cloud thermodynamics and avoid changing too many things at the same time. In addition it would be more realistic. Fortunately, I don't see the results from the sensitivity tests playing a large role. In fact, I also suggest dropping Section 4.3 entirely. As it is now, it doesn't add much to the conclusions, but it also seems to me to contain a lot more than we get to see. Maybe seeds for another paper, rather than "waste" it here?
We have taken the reviewer's suggestion and removed section 4.3.

I also have two more substantial comments. The first relates to the experimental set up in relation to reality and what is explored. In reality, entrainment tends to elevate the top of the cloud layer by mixing in free troposphere air into the boundary layer, and so for a constant cloud-top height, subsidence is required to balance this. Hence, if entrainment is varied while subsidence is kept constant the cloud top height will vary. And they do; this is also what the simulations show and - in my opinion - while realistic it adds two complications.

First, at the end of the manuscript there is a discussion about how the subsidence taking over from entrainment and "collapses" the boundary layer such that the "plume" of aerosols in the free troposphere separates from the boundary-layer top. Now, for the life of me I can't understand why the free troposphere aerosols are not advected along with the PBL top so that the "plume" follows it. These results to me seem unrealistic and I would like that this is properly checked before the manuscript is accepted. I understand that this could lead to a major revision, but I hope not.

The reviewer is correct that this mechanism was not sufficiently explained in the text. The collapse of the boundary layer also has the effect of weakening and thickening the inversion. This effectively increases the distance between cloud top (near the bottom of the inversion) and the tropospheric aerosol (near the top of the inversion). We have updated Fig. 7 to also include the evolution of the temperature inversion for the given cases, and have added discussion of the inversion evolution in the text .

Attached below is the updated Fig 7. The new second row shows that the lower LWP clouds (salt200 and salt600) have a weakening and thickening inversion. The top of this inversion occurs at the same level as the boundary of the free tropospheric aerosol layer. This kickstarts a negative feedback, in which a weaker cloud results in a weakening and thickening inversion, which separates cloud top from the aerosol layer, which in turn continues to weaken the cloud.

[Figure]

Second, if this is realistic it contaminates - if you will - the results. There is the effect of entraining aerosols, which to a first order is related to the aerosol concentration in the free troposphere and, to a second order, has the feedback that a more "active" and persistent cloud, entrains more efficiently thereby contributing to its own persistence. But with the PBL collapse and the separation, which also generates a feedback by additionally suppressing cloud formation, follows yet another aspect that, if realistic, seems to be somewhat artificial. Hence, one could debate if it had been better to tune the different experiments such that the PBL top remained unchanged by changing the aerosols aloft, the same in all runs. I'm not saying this is the right way to go, but it should at least be discussed and this would be the minor revision.

We are not clear exactly what the reviewer is trying to say here. Indeed, a more "active" and persistent cloud will entrain aerosol more efficiently and contributes to its own persistence. In the absence of adequate aerosol to sustain the cloud in the boundary layer, we fail to see what is artificial about a separation of cloud-top from the tropospheric aerosol causing a negative feedback, as described in the response to the prior comment.

Detailed comments:

Figure 1: Maybe plot the height axis scaled to the main inversion base would make things clearer?

It's a little messy regardless of what we do, but we have adopted the reviewer's suggestion. This allowed us to remove the circles that marked the BL top.

Line 64: No black lines in figure, but I guess that would just make if messy? Maybe suggestion above would help?

The black lines are included, we have updated the wording to "The thin black line overlaid on the thicker, colored lines indicates the most likely location of a cloud layer" to be more clear.

Lines 66-67: Suggest "All aerosol profiles except that of 24 July 2020 (Fig. 1b) have higher ... inversion."

Changed as suggested.

Line 70: Drop "can"; you do, don't you?

There is no "can" on line 70 or even near line 70. We assessed all uses of the word "can" in the manuscript and removed two of them.

Lines 107-108: What do you mean by "ice is negligible"? Experience is that much of the precip from even from summer Sc is frozen, rather than drizzle, so in what way is it negligible.

It is negligible insofar as its concentration is low, even for a mixed-phase cloud, and we don't believe that its presence is impacting the qualitative results of our study.

Line 113: Again, change to "Surface heat fluxes were..."; surface moment flux is not zero!!

Fixed as suggested.

Lines 121-122: Inversions "5K per 100 m or stronger" seems to me to be on the strong side, looking at published studies. Its not unusual, but probably not "frequent" either.

The results from MOSAiC as reported in Jozef et al. (2023), which were the focus of this statement, show that this inversion strength is the most common during the summer months.

See Jozef et al. (2023) figure 3: The shallow (greens) and the thick, near-neutral (reds) mixed layers occur at similar frequencies. In both, the > 5K / 100m (dark green, dark red) are the most common.

Lines 152-153: The stable configuration is somewhat unrealistic, for three reasons: 1) Keeping the same surface temperature makes the whole PBL quite warm. It would seem more natural to instead lower the surface temperature and keeping the cloud layer as in control; 2) The structure with a deep quite stable layer is something that I've never seen. Instead the common structure would be two well mixed layers, one associated with the cloud and one with near-surface shear produced turbulence, separated by a shallow inversion; 3) Together, 1) & 2) makes the liquid water content unrealistically high.

Yes, we agree, inclusion of a very shallow mixed-layer at the surface would have made this profile more realistic albeit more complicated. Regardless, we have removed section 4.3 and so there is no need to revise this simulation.

Lines 163: I suggest "the six aerosol sensitivity simulations", since there are more simulations coming.
Changed as suggested.

Lines a63-164: Suggest "appear to attain quasi-steady cloud tops", since this doesn't happen until after some 15h.
Changed as suggested.

Line 171: Do you mean "too small to be observed, or "too small to be measured"? These are not the same you know…
We have removed this descriptor and have kept the sentence to "which is essentially negligible".

Lines 177-179: This seems to contradict earlier statements, and previously published studies by the second author, that near-surface aerosol observations are not representative for the cloud layer
Agreed, in our simulations, despite the lack of surface fluxes, the entire layer from the surface to the cloud top remains well-mixed and as such the droplet and surface aerosol concentrations are similar. Also, as the reviewer has seen, our new simulations no longer show such large droplet concentrations near cloud top. It's not clear to us why that is the case. Regardless, the unrepresentativeness of the surface aerosol concentration is greatest when the cloud layer is decoupled from the surface, as is common in this region, and in such a case, we expect that the second author's previous conclusion would still hold.

Para beginning /w line 203: This needs to be stated carefully. Many of these simulations are essentially optically thin clouds, clouds that are not black bodies, at least large parts of the time. This is especially true for the low aerosol cases. But here these clouds are kept appearing in a well mixed PBL, in part by the initial conditions. In reality, without solar radiation the surface temperature would drop, and the clouds would become decoupled. But the set up here artificially prohibits this development by i) keeping the surface temperature fixed and ii) by the zero heat flux at the surface. One could argue that while "gray clouds" certainly are realistic in the Arctic, such clouds in a (relatively) deep well mixed PBL isn't.
Instead decoupling would occur, possibly with fog forming in the lower layer.
We disagree with the reviewer that the low aerosol case clouds are maintaining a well-mixed PBL. It is evident at hour 15 from the potential temperature profiles in Figure 6d that these cases are developing stable profiles below the cloud layer. This is even more evident at the end of the simulation which we include here. As such, we would argue that these simulations are evolving in the way the reviewer suggests they ought to evolve.

[Figure]

Para beginning /w line 235 & fig6: Variance of vertical velocity most certainly does not go to zero at the surface in reality; is this perhaps only the resolved-scale variance? If so that needs to be clear; scaled with friction velocity, this variance tends towards a fixed rather well-known value.
The reviewer is correct that this is resolved-scale variance. This has been clarified in the text.

Line 237: Drop "average" or rephrase; the average vertical wind is always close to zero, minus the subsidence. It is the strength of updrafts and downdrafts that become stronger or weaker, which is manifested in the variance, but the average vertical velocity remains (essentially) unaffected.
The word "average" was confusing in this case and has been removed.

Line 252: Choice of words; this has nothing to do with "success" or "failure". Maybe "efficiency"?
"Successful" changed to "Efficient".